# Treatment Outcome of 2nd to 5th Metacarpal Fractures: Kirschner Wires Versus Intramedullary Screws

**DOI:** 10.3390/jcm13247626

**Published:** 2024-12-14

**Authors:** Melissa Walde, Dirk Johannes Schaefer, Alexandre Kaempfen

**Affiliations:** Department of Plastic and Hand Surgery, University Hospital of Basel, 4031 Basel, Switzerlandalexandre.kaempfen@usb.ch (A.K.)

**Keywords:** metacarpal fracture, Kirschner wire, intramedullary screw, surgical fixation

## Abstract

**Background/Objectives**: Most metacarpal fractures are isolated, simple, closed, and stable fractures and located distally. They are often caused by accidental falls, strikes by humans, by objects or traffic accidents. The majority can be treated conservatively. When unstable, angulated, malrotated or shortened, a surgical fixation of these frequent fractures is needed. To treat simple, spiral, distal or shaft fractures, intramedullary Kirschner wiring (KW) or intramedullary compression screws (ISs) are used. We wanted to compare the outcomes of those two treatments. **Methods**: In a retrospective study we analyzed the prospectively collected data of our hospital on the indication factors and outcome factors of selected patients with simple or spiral, distal or shaft metacarpal fractures of the second to fifth finger. Indication factors were sex, age, profession, hand dominance, comorbidities, metacarpal finger number, total active range of motion (TAM), rotation, soft tissue damage, localization, articular involvement, fracture type, dislocation and axial shortening. Outcome factors were TAM, rotation, splint time, return to work, bone healing and complications. **Results**: Out of 750 patients, 59 fractures could be included in this study, containing 34 in the KW Group and 25 in the IS Group. Only fracture localization and fracture type were significantly different in the two groups, with more shaft and spiral fractures in the IS Group. The primary outcome of TAM and rotation as well as the secondary outcome of splint time, return to work, bone healing and complication rates showed no significant difference. Only a difference in mean follow-up time was seen. **Conclusions**: Intramedullary screw fixation seems a valid alternative to KW fixation for certain fracture types regarding active range of motion and rotation after treatment, splint time, bone healing and return to work time. Only the tendency of an earlier return to work and a higher rate of full TAM after treatment was seen in favor of intramedullary screws.

## 1. Introduction

Metacarpal fractures account for about 18% of all hand and forearm fractures, constituting 11.7% of all fractures [1,2,3]. Most metacarpal fractures are isolated injuries, characterized by simple, closed, and stable fractures [4]. They often result from accidental falls, direct blows, objects and vehicular accidents. Primarily affecting men between 15 and 24 years, metacarpal fractures [1] frequently involve volarly dislocated subcapital fractures.

The “Arbeitsgemeinschaft für Osteosynthesefragen/Orthopedic Trauma Association” (AO/OTA) classifies metacarpal fractures based on localization into proximal, diaphyseal or distal [5]. Articular involvement is further classified as extraarticular, partial or complete articular, while fracture type is categorized as simple (transverse, oblique or spiral), wedge or multifragmented [6]. Other radiological findings include displacement, axial shortening and rotation. The therapeutic approaches for metacarpal fractures range from conservative to surgical, including Kirschner wiring (KW), intramedullary compression screws (ISs) [7,8], lag screws or plates and screws. The choice of these various surgical approaches is widely discussed and depends on factors such as fracture location, rotation, fragmentation, stability, axial shortening and deformation of the bone as well as patient factors [9,10,11], surgeon’s preference, capabilities and the availability of implants. KW, as well as intramedullary screw fixation, has proved to be reliable for metacarpal neck and shaft fractures [12,13,14]. Although intramedullary screw fixation emerged as a novel technique some years ago, few studies have compared its outcome with other methods [15].

The aim of our study was to retrospectively compare the outcomes of simple and spiral, distal and shaft metacarpal fractures of the second to fifth fingers treated with KW and intramedullary screw fixation.

## 2. Materials and Methods

We retrospectively analyzed all patients with metacarpal fracture of the second to fifth digits between 2016 and 2020 at our University Hospital. Patient demographics, indications for treatment, treatment outcomes and complications were prospectively recorded and noted in the patient charts. Special attention was given to a subgroup analysis of simple distal or shaft metacarpal fracture fixation techniques from the second to fifth fingers, considering this as the most common indication for intramedullary fixation techniques. 

All data were anonymized; no patient informed consent was needed, and ethical approval was obtained from the local ethics committee.

### 2.1. Data

Patients were identified using search terms related to metacarpal fractures in radiology department records. 

Patient demographics, fracture characteristics, treatment modalities and outcomes, including complications, were extracted from charts and X-rays. In detail, patient demographics included sex, age (in years), profession (blue or white collar), hand dominance and comorbidities. 

Fractures were classified according to AO guidelines. Dislocation (in degrees) and axial shortening (in millimeters) were analyzed on X-rays, as shown in Figure 1. Dislocation was measured in lateral view by measuring the angulation of the lines that pass through the center of the metacarpal hand and the shaft [16]. Axial shortening was measured by drawing a line through the most distal point of the heads of two neighboring fingers [17].

Treatment modalities included conservative or operative approaches. Surgical fixations included KW, intramedullary screws, screws only, plates and screws, or complex methods.

By excluding all thumb metacarpal fractures, proximal or fragmented fractures, as well as those managed conservatively or with other surgical techniques such as screws alone, plates and screws, or complex methods, we were able to focus only on patients with metacarpal neck and shaft fractures from the second to fifth digit treated with Kirschner wires or intramedullary screws. Intramedullary screws used in our University Hospital were titanium headless CCS screws (from Medartis AG, Basel, Switzerland), either 3.0 or 2.0 mm, and all patients underwent ergotherapy postoperatively for wound treatment and aftercare with a metacarpal brace (MC brace) for 4 to 6 weeks.

This selective approach was designed to facilitate a precise comparison between the outcomes of intramedullary fixations in a more comparable manner. The primary outcome was total active range of motion and rotation at the last visit, while secondary outcomes included splint time, return to work, bone healing, follow-up time and complications graded according to the Clavien–Dindo classification [18]. The outcomes were observed until completed treatment; patients without complete treatment courses and follow-up were excluded.

### 2.2. Statistics

Data were analyzed using RStudio software version 2022.02.1. Parametric data were compared using unpaired two-sample *t*-tests, while non-parametric data were compared using unpaired two-sample Wilcoxon tests. Categorical outcome data were compared using Fisher’s exact test. Missing data were recoded as “not applicable” (NA) and included in proportion calculations. Significance was set at *p* < 0.05.

## 3. Results

Of an initial 750 patients with metacarpal fractures, 59 patients were included in the study (Figure 2). The KW Group consisted of 34 patients, while the IS Group comprised 25 patients. Both groups included all types of metacarpal fractures except proximal and fragmented fractures. The mean age was 32.9 ± 13.7 years in the KW Group and 26.9 ± 8.7 years in the IS Group. The groups included a majority of men with fractures of the dominant hand, affecting primarily the fifth digit. Fractures in the KW Group were most often located distally (97%) and were transverse (97%), whereas most of the fractures in the IS Group were located at the shaft (72%) and included spiral fractures (8%). This difference was significant between the two groups. Mean dislocation in both groups was 42 degrees; mean axial shortening was 3.4 and 3.6 mm, respectively, in the KW and IS Group (Table 1 and Table 2). 

No significant outcome differences were observed between the two groups for the primary outcomes of total active range of motion and rotation at the last visit, or for difference in splint time or return to work. The KW Group had a significantly longer follow-up time (*p* = 0.008) (Table 3).

Complications for each group are listed in Table 4.

Reduced total active range of motion was seen in five patients in the KW Group and in two patients in the IS Group, as well as malrotation after finished treatment in three patients and one patient, respectively.

Complication rates were similar between the two groups (20.6% and 20.0%, respectively). Bone healing was uneventful and within normal timeframes in all patients in both groups. 

In the KW Group one patient showed a slight Kirschner wire dislocation into the articulation, which led to earlier removal. One developed a *Staphylococcus aureus* infection which needed antibiotic and surgical treatment. One had a synovialitis of the extensor tendon 7 months postoperatively, which led to a synovialectomy. And one patient complained about a painful scar, which improved after scar correction when removing the K-wire. Patients in the IS Group comprised one suffering from tenovaginitis stenosans of the flexor tendon, necessitating intramedullary screw removal and A1. Another patient suffered a secondary dislocation that also led to screw removal and refixation with plates and screws. One patient showed an intramedullary screw with an angulation on a radiographic follow-up check, probably after a secondary trauma. And lastly, one patient complained about a slight skin reaction with hyperkeratosis to the metacarpal brace which was given for protection after surgery.

## 4. Discussions

In this retrospective study, we compared Kirschner wire and intramedullary screw fixation for metacarpal fractures. Both methods showed similar outcomes, with no significant differences in primary or secondary outcomes. Only follow-up time showed a significantly longer observation time in the KW Group. This is explained by the subsequent necessary removal of the Kirschner wires. 

Only a few studies compared the outcomes of metacarpal fractures treated with KW fixation and intramedullary screw fixation. Couciero et al. [19] compared, in a retrospective study, 19 patients treated with intramedullary screws and 11 patients with KW with respect to functional and patient-related outcomes. Similarly to our series, they were not able to detect differences between the two groups for total active range of motion, grip strength, pain, satisfaction or Quick DASH scores. Their main difference was return to work and casting time in favor of the intramedullary screw group. But due to the small sample size, a clear conclusion on any benefits of one particular technique when compared with the other is not possible. In our study we also compared return to work and splint time. Here, we found only a slight tendency without significant difference of earlier return to work in favor of the IS Group. We suggest that shorter intervention time and no need for hardware removal may lead to earlier overall return to work for the intramedullary screw group. Splint time was not different between our groups, since it continued until clinical and radiological union was observed in the 6-week follow-up. 

Another retrospective study by Esteban-Feliu et al. [20] compared a larger sample size of patients (*n* = 253) with displaced short oblique or transverse extraarticular metacarpal or phalangeal fractures treated with plates and screws, KW fixation or intramedullary screws. Their results suggested a better overall outcome for the intramedullary screw group, with a swifter return to work or other regular activities after surgery and fewer secondary procedures for implant removal. In comparison, in our study, we only found a tendency without significant difference in return to work. As in our study, they also found no significant difference in radiographic healing, as well as no difference in TAM. They further observed no difference in time to surgery, follow-up, Quick DASH score or grip strength. We also did not observe any difference in malrotation rates.

Unlike us, another prospective study by Supichyangur et al. [21] found a difference in early TAM. They examined metacarpophalangeal TAM for 12 weeks, comparing headless compression screw and percutaneous KW fixation in 23 metacarpal fractures. They also found an earlier return to work in patients treated with headless compression screws, but no difference in radiographic union at 6 weeks, which supports our results with a tendency of earlier return to work and no difference in bone union.

Compared to the studies above, our study provides more detailed information regarding the outcome of selected metacarpal fractures, including complication rates. We identified a significant difference in fracture type and location between our two groups, which suggests the possibility of a certain indication bias. KW fixation was preferably performed in distal metacarpal fractures, whereas spiral shaft fractures were fixed using intramedullary screws. This may be attributed to a learning curve with intramedullary implants during the study time, as their use has only been promoted since around 2015 at our institution. A larger sample size could have potentially reduced this effect. However, the retrospective nature of this study and the lack of randomization have their limitations. To exclude selection bias, a randomized controlled trial would be necessary, which would raise methodological questions, especially regarding the necessity for a secondary intervention in the KW Group. Also, return to work following primary treatment may have been influenced by a larger proportion of white collar workers in the IS Group. 

In our literature research on the theory and mechanisms behind the fixation methods we compared, we discovered a biomechanical study conducted by Avery et al. [22]. Their research found headless compression screws for metacarpal neck fractures to show superior results in terms of load to failure, 3-point bending and axial loading when compared to Kirschner wires. Additionally, the biomechanical comparison of intramedullary screw fixation, dorsal plating and Kirschner wire fixation by Galbraith et al. [23] showed headless compression screws to be superior to Kirschner wires in terms of bending and torsion. However, they were found to be less stable than dorsal plating. The initial stability provided by Kirschner wires seemed sufficient to handle the expected loads during the early rehabilitation period, while dorsal plates and intramedullary compression screws provided stability far in excess.

Confirming this theory, intramedullary screw fixation, as well as KW fixation proved to be valid treatments for unstable metacarpal fractures [17]. The long-established KW fixation allows early active motion, but requires removal due to loosening and migration of the wires, with potential extensor tendon injury [24,25], as seen in one of our cases. 

The novel use of intramedullary screws offers a reliable minimally invasive approach for fixation of metacarpal fractures [26,27].

In a study of 160 metacarpal fractures treated with intramedullary screws by Warrender W et al., only four complications (2.5%) were described [27]. This included one case of nickel allergy, one a broken screw after repeat trauma, and two bent intramedullary screws, with no serious complications noted. In our series, we encountered one bent intramedullary screw, but it did not result in functional deficit. In another study involving 20 patients with a minimum 3-month follow-up after intramedullary screw fixation, two complications of shaft re-fracture from blunt trauma were reported, along with one occasional clicking during metacarpophalangeal joint motion; however, no radiographic osteoarthritis was observed at the latest follow-up [24]. In our cases, we did not observe any re-fracture, although we did note a secondary dislocation with malrotation and one case of tenovaginitis stenosans 3 months postoperatively, both of which were treated without complications.

Further effects on cartilage and extensor tendon defects were examined by Urbanschitz L et al. [28] on cadaveric hands. The median articular joint surface defect in mm^2^ was 7–9 (5–6% of the articular surface); the tendon damage was <25% for the retrograde mini-open method and a maximum of 25–49% for the retrograde percutaneous method. The clinical relevance for development of osteoarthritis is unknown and has not been described so far. In reasonable agreement with Urbanschitz et al. [28] are also the three-dimensional data of ten Berg et al. [29]. The mean articular surface area of the metacarpal head used by the countersunk headless compression screw thread was 12%, 8% and 4%, respectively, in neutral position, coronal plane arc and sagittal plane arc. They concluded a minimal occupation of the head surface and subchondral volume occupied by the headless compression screw. Articular surface area violation was least during the more clinically relevant sagittal plane arc of motion. However, further studies with a long follow-up are needed to detect clinical relevance in the development of osteoarthritis. In our study with a mean follow-up time of 3 months, no osteoarthritis was observed. 

Our study allowed us to evaluate the early outcomes in the more novel treatment of intramedullary screw fixation in comparison to the longer established KW fixation. No significant differences were shown, only a tendency of earlier return to work and a similar complication rate.

## 5. Conclusions

In conclusion, no significant difference in primary or secondary treatment outcome was found in our study comparing KW and intramedullary screw fixation in metacarpal fixation of the 2nd to 5th metacarpal. A tendency of earlier return to work and a higher rate of full TAM after treatment with an intramedullary screw and fewer complications was noted but not significant.

## Figures and Tables

**Figure 1 jcm-13-07626-f001:**
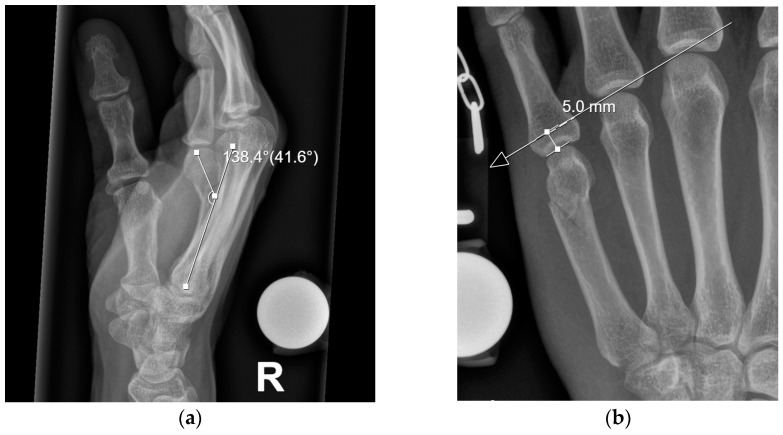
(**a**) Example of the measurement of dislocation; (**b**) example of the measurement of axial shortening.

**Figure 2 jcm-13-07626-f002:**
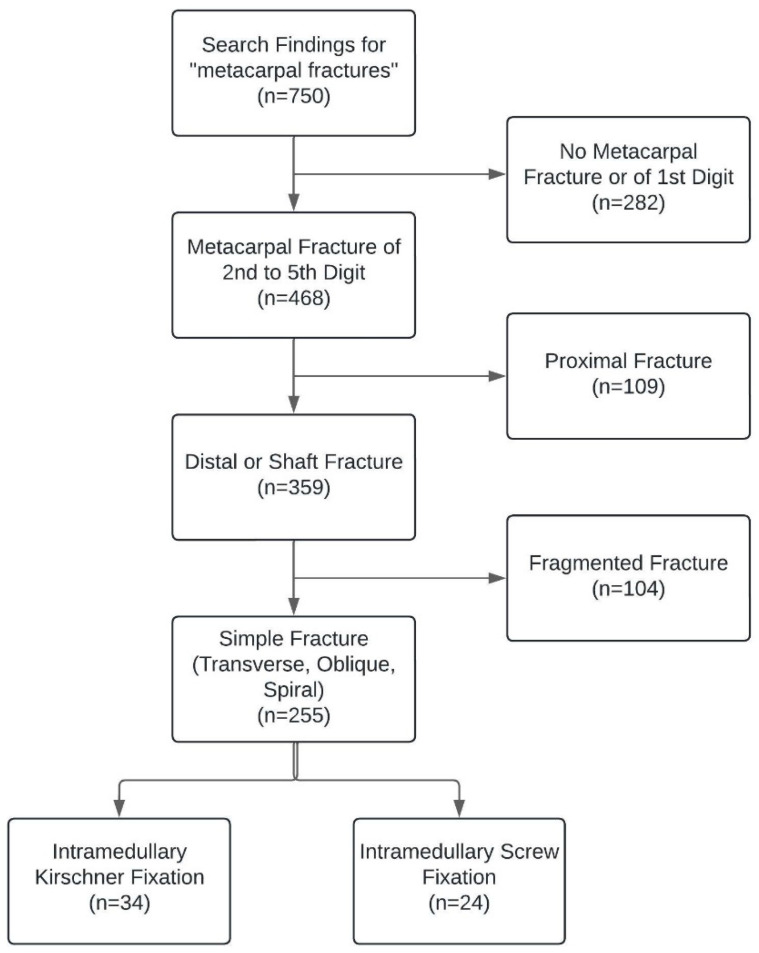
Flowchart of the selection process.

**Table 1 jcm-13-07626-t001:** Demographics.

	KW Fixation *n* = 34	Intramedullary Screw Fixation *n* = 25	*p*-Value
**Age** (**mean years ± SD**)	32.9 ± 13.7	26.9 ± 8.7	0.0641
**Gender**			
female	3 (8.8%)	2 (8.0%)	1
male	31 (91.2%)	23 (92.0%)	
**Hand dominance**			
NA	1 (2.9%)	1 (4.0%)	1
adominant	5 (14.7%)	4 (16.0%)	
dominant	28 (82.4%)	20 (80.0%)	
**Profession**			
NA	10 (29.4%)	2 (8.0%)	0.0845
blue collar	15 (44.1%)	17 (68.0%)	
white collar	9 (26.5%)	6 (24.0%)	

**Table 2 jcm-13-07626-t002:** Fracture characteristics.

	KW Fixation *n* = 34	Intramedullary Screw Fixation *n* = 25	*p*-Value
**Metacarpal number**			
2	0 (0.0%)	0 (0.0%)	0.1486
3	1 (2.9%)	2 (8.0%)	
4	1 (2.9%)	4 (16.0%)	
5	32 (94.1%)	19 (76.0%)	
**Total active range of motion before treatment**			
NA	3 (8.8%)	1 (4.0%)	0.8232
full	7 (20.6%)	6 (24.0%)	
reduced	24 (70.4%)	18 (72.0%)	
**Rotation before treatment**			
NA	3 (8.8%)	0 (0.0%)	0.272
no	11 (32.4%)	11 (44.0%)	
yes	20 (58.8%)	14 (56.0%)	
**Soft tissue damage**			
NA	0 (0.0%)	0 (0.0%)	1
no	30 (88.2%)	23 (92.0%)	
yes	4 (11.8%)	2 (8.0%)	
**Localization**			
proximal	0 (0.0%)	0 (0.0%)	**<0.0001**
shaft	1 (2.9%)	18 (72.0%)	
distal	33 (97.1%)	7 (28.0%)	
**Articular involvement**			
extraarticular	33 (97.1%)	25 (100.0%)	1
intraarticular	1 (2.9%)	0 (0.0%)	
**Fracture type**			
fragmented	0 (0.0%)	0 (0.0%)	**0.0058**
oblique	0 (0.0%)	5 (20.0%)	
transverse	33 (97.1%)	18 (72.0%)	
spiral	1 (2.9%)	2 (8.0%)	
**Dislocation** (**mean degrees ± SD**)	43.1 ± 14.1	41.8 ± 12.5	0.722
**Axial shortening** (**mean millimeters ± SD**)	3.4 ± 1.4	3.6 ± 1.2	0.6888

**Table 3 jcm-13-07626-t003:** Characteristics and comparison of outcome measures.

	KW Fixation (*n* = 34)	Intramedullary Screw Fixation (*n* = 25)	*p*-Value
**Splint time** (**mean weeks ± SD**)	6.1 ± 0.5	5.9 ± 1.0	0.4346
**Total active range of motion after treatment**			
NA	6 (17.6%)	2 (8.0%)	0.3796
full	23 (67.6%)	21 (84.0%)	
reduced	5 (14.7%)	2 (8.0%)	
**Rotation**			
NA	5 (14.7%)	3 (12.0%)	0.795
no	26 (76.5%)	21 (84.0%)	
yes	3 (8.8%)	1 (4.0%)	
**Bone healing**			
NA	8 (23.5%)	5 (20.0%)	0.5228
early	24 (70.6%)	20 (80.0%)	
medium	2 (5.9%)	0 (0.0%)	
late	0 (0.0%)	0 (0.0%)	
**Return to work** (**mean days ± SD**)	69.1 ± 27.1	62.2 ± 44.7	0.0699
**Follow-up** (**mean days ± SD**)	108.5 ± 102.5	67.8 ± 47.9	**0.0082**

**Table 4 jcm-13-07626-t004:** Complications for each group.

Complications	KW Fixation (*n* = 34)	Intramedullary Screw Fixation (*n* = 25)
Clavien–Dindo Classification		
Grade I	1	2
Grade II	0	0
Grade IIIa	3	2
Grade IIIb	0	0
Malrotation	3	1
Bone healing late/non-union	0	0
**Total**	**7** (**20.6%**)	**5** (**20.0%**)

## Data Availability

The data are unavailable due to ethical restrictions.

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
