# Peer review of "Treatment Outcome of 2nd to 5th Metacarpal Fractures: Kirschner Wires Versus Intramedullary Screws"

_jcm, 2024, doi:10.3390/jcm13247626_

Round 1

Reviewer 1 Report

Comments and Suggestions for Authors

Dear Authors, 

The subject of the manuscript is interesting adequate introduction but can be improved with data from the literature on how to establish the indication for the use of one or another of the studied surgical techniques.

In terms of material and method, I think that the criteria that led to the exclusion of some patients from the study should be mentioned the criteria by which the surgical technique was established and the choice of material should be mentioned is osteosynthesis

It should be mentioned if the patients' informed consent was obtained

It should be mentioned that the study also included the evaluation of complications the time interval at which the results were evaluated, the evaluation scales of the results that were used patients required physical therapy and if this was performed

Thank you

Best wishes

Reviewer 2 Report

Comments and Suggestions for Authors

Thank you to the authors for this interesting study. The manuscript is well-structured, with a correct methodological approach.

The results and conclusions are coherent and appropriately aligned. The discussion is well-structured, and the comparisons relevant.

The English is generally good, but some parts require revision (e.g., lines 190 and 211).

Additionally, could the authors explain why the Clavien-Dindo classification was chosen? Are there other classifications that might be more appropriate for this context?
